# OpenReview forum: "QuadSentinel: Sequent Safety for Machine-Checkable Control in Multi-agent Systems"
_ICML.cc/2026/Conference — Submitted to ICML 2026_

### Official Review · Reviewer_dLc1 · 2026-03-09

**Soundness:** 3
**Presentation:** 2
**Significance:** 2
**Originality:** 2
**Overall Recommendation:** 4
**Confidence:** 4

**Summary:**

This work compiles policies into machine-checkable rules, and propose  QUADSENTINEL, a four-agent guard (state tracker, policy verifier,
threat watcher, and referee). The system updates predicates and states during execution, verifies whether policy violations occur, and finally makes safety judgments. Experiments on ST-WebAgentBench and AgentHarm demonstrate state-of-the-art performance.

**Compliance With Llm Reviewing Policy:**

Affirmed.

**Final Justification:**

The rebuttal addressed my main concerns.

**Key Questions For Authors:**

Question 1: For the message-level guardrail, what types of policies did the authors use? Are these policies limited to prompt injection rules, or do they also cover other comprehensive categories?

Question 2: In the runtime efficiency analysis, the authors claim that QuadSentinel is more efficient than ShieldAgent. However, QuadSentinel is a multi-LLM system and involves additional LLM generation steps. Could the authors provide an intuitive explanation of why QuadSentinel achieves better efficiency than ShieldAgent?

Question 3: How does the system handle future risks that fall outside the scope of the current policies? What is the generalization capability of the proposed method in such scenarios?

**Limitations:**

Yes

**Strengths And Weaknesses:**

Strengths:

The proposed four-agent guard (state tracker, policy verifier, threat watcher, and referee) is a practical design for shielding unsafe operations and messages through state tracking. The experimental results on two benchmarks demonstrate the effectiveness of the proposed method.

Weaknesses:

The idea of utilizing verifiable policies for shielding is not entirely novel. Although the paper claims that the method can provide additional defense at the message-transfer layer in multi-agent systems, the experimental results do not clearly demonstrate this advantage. Also, the paper contains overly polished wording, which makes some parts difficult to follow.

---

> ### Author Rebuttal · Authors · 2026-03-31
>
> We thank the reviewer for the thoughtful feedback and for recognizing both the practical design of QuadSentinel (QS) and its empirical effectiveness.
>
> _1. Novelty._
>
> At a broad level, we agree that "compile policy into checkable rules" is not new. This is a deployment-oriented contribution, not a new logic.
>
> Compiling natural-language policies into machine-checkable rules is a prerequisite for rigorous runtime enforcement; **our contribution is what QS does with those rules in a multi-agent runtime.**
>
> Our novelty claim is architectural rather than policy representation alone. Specifically, QS combines:
>
> a. guarding a multi-agent system rather than a single-agent setting;
>
> b. **explicit runtime enforcement over both inter-agent messages and environment actions;**
>
> c. specialized guard roles with persistent state and risk-adaptive adjudication; and
>
> d. **a low-overhead design that enables practical runtime monitoring.**
>
> The current evaluation is **end-to-end rather than component-isolating**: it demonstrates the effectiveness of the full guard architecture, but does not separately quantify the marginal contribution of message-level enforcement. Our claim is thus **architectural and end-to-end**, not that the current experiments provide a standalone measurement for the message channel in isolation. At the same time, the evaluation does exercise **coordination-layer risks through multi-agent orchestration**, which is why we believe it remains informative for this setting.
>
> These benchmarks were chosen for apples-to-apples comparison with prior guardrails under the same tasks, policies, and block/allow protocol. They support an end-to-end evaluation of the full QS architecture, but not a clean isolation of message-level guarding alone. We will make that scope explicit in the revision.
>
> Taken together, these architectural choices **go beyond the use of verifiable policies in isolation.** We will revise the text to make this distinction more explicit.
>
> _2. What policies are used at the message level?_
>
> The message-level mechanism is not restricted to prompt injection; it applies whenever a policy can be expressed in terms of observable properties of an inter-agent message, such as message content, sender/recipient context, or instruction-bearing content.
>
> This matters because the guard can intervene at the coordination layer, before unsafe instructions or policy-violating content are propagated into downstream actions.
>
> In the current experiments and illustrated examples, prompt injection/instruction manipulation is the clearest message-borne category we explicitly study.
>
> We will revise the text to make this **evaluated scope** explicit and to distinguish mechanism scope from current empirical coverage.
>
> _3. Why is QS more efficient than other guardrails?_
>
> The key reason is the **cost structure of the checks rather than the raw number of components or steps.**
>
> The online checks are narrow and low-latency rather than full free-form generations. The state tracker and threat watcher mainly produce short predicate/risk updates with a lightweight model (e.g., GPT-4o-mini), while the referee uses hierarchical escalation, so only deny or high-ambiguity cases are sent to a stronger model (e.g., GPT-4o). This lets the system **reserve expensive deliberation for the minority of ambiguous cases** rather than pay that cost on every interaction.
>
> In addition, the state tracker updates only a **small candidate predicate set** via top-k retrieval plus a must-check set, rather than exhaustively reevaluating all predicates or generating runtime code.
>
> In practice, this is why QS shows **lower runtime overhead than ShieldAgent and GuardAgent** in our runtime efficiency analysis.
>
> We will make this cost argument more explicit in the revision.
>
> _4. Generalization to future risks outside current policies._
>
> More broadly, this is **a general boundary of policy-grounded safety systems rather than a limitation unique to QS**: if a risk concept is not represented in the current policy book, no rule-based guard can guarantee its detection.
>
> This is the same basic tradeoff seen in signature- or rule-based security systems such as anti-virus: they are strong on known, codified patterns, but new risks must first be surfaced and then added to the rule base.
>
> So, our design prioritizes **reliable, auditable enforcement of explicitly registered policies** rather than asking an LLM to speculate about unbounded future risks on the fly.
>
> For risks outside the current policy book, the Threat Watcher serves as a **behavioral backstop**: recurrent anomalous behavior can raise threat levels, trigger stricter referee thresholds, or prompt manual review. Newly identified risks can then be **compiled offline into additional predicates and rules for subsequent deployments.** This mechanism is heuristic and **not designed to provide complete coverage of uncodified risks.**
>
> We will state this tradeoff and extension path more explicitly in the revision.

---

> > ### Author Rebuttal · Reviewer_dLc1 · 2026-04-03
> >
> > Thanks for the clarification. I raise my score to 4. Please also take the other reviewers’ suggestions into consideration in the revision.

---

### Official Review · Reviewer_UzqV · 2026-03-13

**Soundness:** 3
**Presentation:** 3
**Significance:** 2
**Originality:** 3
**Overall Recommendation:** 3
**Confidence:** 4

**Summary:**

This paper proposes a multi-agent system, quadSentinel, to guard the agentic system from attack. Specifically, quadsentinel can employ machine-checkable control to verify different kinds of violations in the agentic system. The experiment results show that quadSentinel can outperform the current guard agents and achieve the best performance.

**Compliance With Llm Reviewing Policy:**

Affirmed.

**Final Justification:**

Thank you to the authors for the clarification. I agree that the policy-to-rule translation is not the central technical contribution of the paper, although it remains a foundational component of the overall pipeline. I also acknowledge the value of the online verification stage based on machine-checkable predicates and rules. However, because the effectiveness of the entire framework ultimately depends on the quality of the initial policy translation, and I do not yet see a convincing solution to this limitation, I remain uncertain about the practical robustness of the full pipeline. For instance, ShieldAgent also relies on an automatic policy translation step, suggesting that this is a broader challenge for this line of work rather than an isolated design choice. Overall, I will maintain my score but I would not argue for rejection because of this concern.

**Key Questions For Authors:**

1. When applying QuadSentinel or other guard agent systems, which one of recall or precision is more important?
2. Can you further show the cost of the rule translation steps?

**Limitations:**

yes

**Strengths And Weaknesses:**

**Strengths**
- the paper is well written and easy to follow.
- The paper is addressing an important problem to guard the agentic system in runtime. Overall, with the rise of llm-backboned agents, having explicit functionality to guard the system, this is an important problem.

**Weaknesses**
- As also stated in the limitations, the policy-to-rule translation is the most challenging part of the entire end-to-end pipeline. Although the author states that this is an offline process, I believe that having human validators translate the policy into a rule is not a method that can be applied to every agentic system. This somehow limits the generalizability of these methods.
- Additionally, the authors also mentioned human-in-the-loop validation, but I think this paper should provide more details about the process. Specifically, they should clarify: 1. How many experts did they employ and for how long? 2. How many human hours were put in for the validation? 3. What was the level of agreement between different human labelers? These details are the foundation of this work, and I think the paper should discuss them more.
- In line 129, while the verification relies on explicit logical sequents over boolean predicates, the predicates are retrieved from candidates' pool with an embedding similarity metric. How can we ensure all relevant predicates are retrieved? And a counter-intuitive results in table 4 make it even more difficult to understand the role of predicates in the whole pipeline. 100% of all predicates do not make the performance better. Can the author provide more justification?

Overall, I am mostly concerned with the generalization of the quadsentinel, with the need to have human experts engaging in the translation steps. Is it difficult to apply QuadSentinel in a new task?

---

> ### Author Rebuttal · Authors · 2026-03-31
>
> We thank the reviewer for recognizing the importance of runtime guarding for agentic systems and for the positive assessment of the paper's clarity. Below we address the remaining concerns on generalizability, predicate coverage, and the recall-precision tradeoff.
>
> Our claim is not end-to-end formal verification of arbitrary agent systems.
> The contribution is a practical runtime guard for software LLM-agent systems that
>
> (i) compiles natural-language policies once offline into machine-checkable predicates and rules,
>
> (ii) guards both inter-agent messages and environment actions online, and
>
> (iii) achieves a better safety/latency tradeoff than prior baselines on two complementary benchmarks.
>
> The stochastic boundary is predicate extraction; the decision layer over the resulting predicate store is explicit and auditable.
>
> ***1. Generalizability, offline policy registration, and human review.***
>
> In our pipeline, policies are compiled **offline** by the LLM, while human-in-the-loop review serves as an **optional** pre-deployment validation/refinement step rather than a runtime requirement or a manual translation process from scratch for each new domain. Policy-to-rule compilation is a real practical consideration, so the key question is whether its burden is manageable. In our current setup, this burden was **modest and entirely offline**: for the 38-rule policy set, LLM-based translation used about 46k tokens and 529 seconds in total, and one domain expert reviewed the compiled rules for each benchmark in under an hour.
>
> In the current evaluation, human review is a single-expert pre-deployment validation pass. We report it to quantify onboarding cost, not to claim multi-annotator agreement evidence. The practical question is therefore not whether QuadSentinel eliminates policy registration, but whether that one-time registration burden is manageable for deployment; in our setup, it was modest and entirely offline. More broadly, this step is not unique to QuadSentinel: any method that aims to enforce natural-language safety policies through machine-checkable control must first convert those policies into an executable representation.
>
> QuadSentinel's contribution is to make this practical for multi-agent runtime guarding: policies are compiled once offline into predicates and rules, optionally reviewed before deployment, and then enforced online with **low overhead** over both inter-agent messages and environment actions. In practice, applying QuadSentinel to a new task requires a policy book and a **one-time offline policy registration step**, while the **runtime guard architecture** remains reusable across agentic systems.
>
> ***2. Predicate coverage and the Table 4 result.***
>
> Predicate coverage is **not based on embedding similarity alone**. At runtime, the candidate set is the union of a top-k retrieved set and a deterministic must-check set derived from rule dependencies, so predicates required by known tool/rule dependencies are included even when they are not highly ranked by semantic similarity. In addition, when confidence is low, the verifier can request state expansion before issuing a verdict.
>
> Table 4 should be read in that context. The **k = 100%** setting means that the state tracker forces updates for all predicates at every step; it does not give the system more policy knowledge. Most predicates are irrelevant to any given interaction, so updating all of them mainly injects noise into downstream rule checking rather than improving useful coverage.
>
> This is exactly what the ablation shows: moving from k = 10% to k = 100% leaves recall unchanged (85.2 to 85.2) but lowers precision (97.4 to 92.0) and increases the false-positive rate (2.3 to 7.4). Thus, the top-k module is **not only an efficiency optimization; it is also a robustness mechanism** that preserves necessary predicates while filtering irrelevant ones.
>
> ***3. Recall versus precision.***
>
> For runtime safety guards, recall is the primary objective because false negatives correspond to unsafe actions that go undetected. At the same time, precision is critical for practical deployment, since excessive false positives can block benign actions and make the guarded system unusable. QuadSentinel is designed to **improve this tradeoff rather than optimize either metric in isolation**, which is why we report both recall and false-positive behavior. In our results, the goal is not merely high recall, but high recall together with a low false-positive rate, so that safety gains do not come at the cost of making the protected system impractical.

---

> > ### Author Rebuttal · Reviewer_UzqV · 2026-04-01
> >
> > My main concern remains unresolved. The rebuttal reframes policy-to-rule translation as a one-time offline onboarding cost, but my objection was not primarily about cost. This step is the semantic foundation of the whole method: all downstream machine-checkable verification only holds if the translated predicates/rules are themselves correct and complete. Reporting 46k tokens, 529 seconds, and a single expert review in under an hour does not establish validity and reliability. In fact, the rebuttal explicitly confirms that the current evaluation relies on a single-expert pre-deployment validation pass and does not provide multi-annotator agreement evidence. This makes it difficult to assess whether the rule set is reproducible, whether different experts would derive similar executable policies, and how much the final performance depends on expert intervention. While the authors argue that such translation is not unique to their method, this does not address the issue: in this work, the translated rules directly define the operational semantics of the guard. Therefore, this step cannot be treated as a minor offline detail. It is the foundation of the system, and its correctness must be rigorously evaluated.

---

> > > ### Author Response · Authors · 2026-04-05
> > >
> > > Thank you for the follow-up. We agree that the current paper does not establish reproducibility or expert-independent validity of the offline policy-registration stage, and we should narrow our wording accordingly.
> > >
> > > The key point is the scope of what the current experiments evaluate. The reported results are a **comparative runtime evaluation under fixed policy specifications**, not a validation of expert-independent policy registration. In those experiments, the policy sources are benchmark-grounded and shared rather than unique to QuadSentinel: ST-WebAgentBench uses benchmark-provided safety rules, AgentHarm uses the official Reddit User Rules also used by prior work, and the closest baseline, ShieldAgent, also operationalizes natural-language policies into executable rules for runtime enforcement.
> > >
> > > Under that framing, the offline-specification issue is real, but it is **not uniquely disqualifying for this experimental comparison**. What it limits are claims about reproducibility across experts, onboarding to new domains, and expert-independent policy registration. What it does **not** invalidate is the narrower result established by the current experiments: **once a policy book is fixed**, QuadSentinel's runtime architecture enforces those policies over both inter-agent messages and environment actions, and yields the reported comparative safety/latency tradeoff.
> > >
> > > To directly address your question about dependence on expert intervention in our current setup, we also evaluated a no-human-review variant on AgentHarm (GPT-4o, Magentic-One), using the automatically compiled policy book without expert review. This variant achieved
> > >
> > > 90.3 Acc. / 93.3 Prec. / 86.9 Rec. / 6.3 FPR, compared with
> > >
> > > 91.5 Acc. / 97.4 Prec. / 85.2 Rec. / 2.3 FPR for the expert-reviewed version.
> > >
> > >
> > > We do **not** view this as validating the offline stage. Rather, it isolates one part of the concern: in our current setup, expert review mainly improves calibration and false-positive control, while the runtime architecture still preserves most of the end-to-end effectiveness without expert review. So the reported runtime gains are not explained solely by expert intervention.
> > >
> > > Accordingly, the current evidence supports a narrower claim: under the benchmark-grounded policy specifications used in the evaluation, QuadSentinel improves the runtime safety/latency tradeoff relative to prior guards, while broader questions about the reproducibility of policy registration remain open.
> > >
> > > We will revise the paper accordingly and remove stronger wording that could suggest the offline registration stage has been fully validated. We agree that broader validation of that stage remains an important limitation. Our claim is the narrower one supported by the current evidence: a meaningful comparison of runtime guard architectures under shared, benchmark-grounded policy specifications.

---

### Official Review · Reviewer_HSrt · 2026-03-16

**Soundness:** 3
**Presentation:** 2
**Significance:** 3
**Originality:** 3
**Overall Recommendation:** 4
**Confidence:** 5

**Summary:**

The paper describes an LLM-based multi-agent guard system (QuadSentinel) for runtime, execution-trace-based checking of safety properties defined as propositional formulas over boolean predicates on a labeled state transition model of the analyzed multi-agent system. The term “multi‑agent system” is used in an architectural sense (multiple LLM components exchanging messages), which differs substantially from the classical MAS literature on distributed optimization and cooperative control, where autonomous agents implement and are validated against policies optimizing a shared quantified objective. QuadSentinel only checks the actions and messages taken at runtime; there are no guarantees of the absence of safety violations in the underlying model because checking is runtime-only and uses probabilistic LLM components to infer predicate truth and support property evaluation. A key contribution of the approach is reducing runtime monitoring overhead compared to analogous systems, achieving about 0.33× time overhead versus ShieldAgent (~0.62×) and GuardAgent (~1.24×), while also improving safety metrics. GPT‑4o and GPT‑4o‑mini are used to translate policies into predicates and rules, infer predicate truth values, compute risk scores, and help adjudicate rule violations, but they are not specially fine-tuned for this logical checking task; instead, end-to-end system performance is evaluated on ST‑WebAgentBench and AgentHarm, where on AgentHarm QuadSentinel attains 91.5% accuracy, 97.4% precision, 85.2% recall, and 2.3% FPR, outperforming ShieldAgent and GuardAgent.

**Compliance With Llm Reviewing Policy:**

Affirmed.

**Key Questions For Authors:**

Can you please explain why you use sequent terminology while the analysis implemented looks like semantic entailment ?
The use of the sequent turnstile Γ_t ⊢ ψ is misleading given how the guard is implemented. The paper discusses sequent safety, yet at runtime the system does not construct syntactic sequent proofs in the sense of standard sequent calculi (e.g., as in "Logic in Computer Science" by Huth & Ryan and many other similar texts), but instead evaluates whether the propositional rule formula is satisfied under the concrete valuation induced by Γ_t. This is closer to semantic entailment or model checking, and would be more naturally written as Γ_t ⊨ ψ, or explicitly described as using ⊢ in a purely semantic sense.

**Limitations:**

I would like to see explicit discussion of the fact that checking safety properties is stochastic and the impact of this limitation on the problem domain (it would not be admissible in a mission critical system).

**Strengths And Weaknesses:**

The approach is sound. The presentation can be somewhat improved by summarizing early on the system's purpose and verification mechanism similar to the summary in this review. It took a bit of time to understand the verification algorithm (i.e. what is actually doing the verification, as turned out it was GPT-4o, not a traditional program analysis algorithm implementation)
The paper addresses a significant and important problem of runtime verification of LLM based multi-agent systems.
The approach conceptually builds on existing systems, ShieldAgent and GuardAgent. It manages to reduce the monitoring overhead by  compiling policies into predicates over observable states/events as propositional formulas and maintaining a global predicate store. Quad Sentinel is itself a multi-agent system.
The strength of the system in that it does reduce overhead compared to its main competitors.
One of the main weaknesses is that checking of predicates is stochastic. And even assuming the stochastic result on safety property is admissible (which would not be in a mission critical software system, e.g. I would not want to ride in an elevator whose safety property of not moving with its doors open is only 97% accurate) then some additional fine-tuning and evaluation should be done on the LLMs used for that checking.

---

> ### Author Rebuttal · Authors · 2026-03-31
>
> We appreciate the reviewer's positive assessment and helpful suggestions. We also appreciate the reviewer's careful attention to the distinction between the symbolic checking layer and the LLM-based components that support predicate extraction and runtime monitoring.
>
> More precisely, **the intended guarantee is the following**: given the current predicate valuation, **rule evaluation is deterministic, machine-checkable, and auditable**; the stochastic part lies in the LLM-mediated perception layer that maps execution traces to predicate values. Accordingly, **end-to-end safety is bounded by predicate extraction accuracy and policy coverage**.
>
> We will revise the paper to make this boundary more explicit and to state more clearly that **QuadSentinel is a practical runtime guardrail for software LLM-agent systems**, rather than **a certified verifier for mission-critical domains requiring end-to-end formal assurance**. This is also the sense in which we position QuadSentinel: as an explicit and auditable runtime defense layer that reduces risk in LLM-agent execution, not as a substitute for full formal assurance in high-assurance control settings.
>
> Additional fine-tuning may further improve predicate extraction, but **it is not a prerequisite to our claim**. Our contribution is that prompt-engineered predicate extraction paired with deterministic rule evaluation already yields the reported safety and overhead gains without task-specific fine-tuning. We will clarify this scope in the revision so that the contribution is not read as depending on a specialized predicate-checking model.
>
> At runtime, **the verifier performs valuation-based semantic checking** over the current predicate store, i.e., it checks whether the rule formula is satisfied under the concrete valuation induced by $\Gamma_t$, rather than syntactic proof search in a sequent calculus. We agree that **the current sequent terminology is too strong for the implemented verifier**.
>
> We will therefore revise the notation and technical framing accordingly, e.g., replacing $\Gamma_t \vdash \psi$ with $\Gamma_t \models \psi$, and clarifying that the verifier returns violated rules together with the supporting predicates. We will also strengthen the limitations discussion accordingly, explicitly stating that QuadSentinel's end-to-end effectiveness is limited by stochastic predicate extraction and policy coverage, even though the downstream rule-evaluation layer is explicit and auditable.

---

> > ### Author Rebuttal · Reviewer_HSrt · 2026-04-04
> >
> > Thank you for clarifying the mechanism of rule evaluation, predicate extraction and the focus of the contribution. Thank you for addressing the terminological issue regarding semantic entailment.

---

### Official Review · Reviewer_Zamx · 2026-03-21

**Soundness:** 2
**Presentation:** 3
**Significance:** 2
**Originality:** 3
**Overall Recommendation:** 3
**Confidence:** 3

**Summary:**

This work highlights safety concerns in multi-agent systems, where risks can arise from inputs, the environment, and agent behaviors. The authors then propose a guardrail system, QuadSentinel, that monitors multi-agent systems and ensures runtime safety compliance through machine-checkable control. The QuadSentinel system consists of four independent safety agents (state tracker, threat watcher, policy verifier, and referee) that collaborate to observe, verify, and regulate the safety of the target multi-agent system.

**Compliance With Llm Reviewing Policy:**

Affirmed.

**Key Questions For Authors:**

- Does QuadSentinel apply to each intermediate output of the multi-agent system?
- Is the policy book static once compiled? Would it be possible to update it dynamically during runtime?
- Can components interact bidirectionally to enable error correction, or is the flow strictly unidirectional?

**Limitations:**

Yes.

**Strengths And Weaknesses:**

Strengths:
- The proposed system studies runtime safety concerns in multi-agent systems, which is a real-world important problem.
- From the experimental results, the proposed framework significantly reduces the false positive rate while maintaining high recall.

Weaknesses:
- The multi-agent guarding approach can introduce significant cost and latency overhead due to the complex component design, which may limit its practicality.
- The robustness of policy translation is unclear, which could be a major issue for the downstream pipeline. Although the authors claim that this step can be done offline and include human-in-the-loop verification, it requires additional effort and is not well adapted to the overall multi-agent guarding system.
- The experiments are conducted on only two benchmarks, while there are many related agent benchmarks that are not included. I would suggest the authors include more of them.
- It seems that different components operate sequentially rather than interacting dynamically with each other. This means later components depend on the outputs of earlier ones, making the system less reversible and less flexible for automatic correction.

---

> ### Author Rebuttal · Authors · 2026-03-31
>
> We appreciate the reviewer's recognition that runtime safety in multi-agent systems is an important problem and that QuadSentinel (QS) maintains high recall while significantly reducing false positives.
>
> We didn't claim end-to-end formal verification. Also see rebuttal to UzqV.
>
> _1. Practicality under multi-agent guarding._
>
> As Zamx notes, making a multi-agent guard practical without incurring high cost and latency overhead is challenging. QS is designed precisely around that constraint. The runtime results don't support the concern that multi-agent guarding introduces prohibitive cost or latency. On AgentHarm, it adds 0.33x time overhead, compared with 0.62x for ShieldAgent and 1.24x for GuardAgent. This comes from selective predicate updates and staged adjudication: the State Tracker evaluates only Top-k U Must-Check predicates, and the Referee escalates only for deny or ambiguous cases rather than triggering full re-analysis at every step.
>
> _2. Offline specification vs. online enforcement._
>
> Policy-to-rule translation is an **offline** specification step, not part of the online guard loop. This is deliberate: runtime enforcement should operate over an explicit **machine-checkable rulebook** rather than repeatedly interpreting ambiguous natural-language policy text on the fly.
>
> This's a **one-time offline registration step**, not manual rule authoring from scratch for each deployment. Human review is used only for pre-deployment validation/refinement. In our evaluated settings, the burden was modest: for each benchmark, **one domain expert** reviewed the compiled rules in **under an hour**, and for the **38-rule** policy set, the offline translation itself consumed about **46k tokens** and **529 seconds** in total.
>
> Applying QS to a new domain, therefore, requires a policy book and a one-time registration step, but **not retraining or modifying the protected agent system**.
>
> _3. Evaluation beyond a single narrow setting._
>
> The current evaluation is limited but **not single-regime**. We chose **ST-WebAgentBench** and **AgentHarm** because they are complementary rather than redundant: **ST-WebAgentBench** emphasizes policy-rich enterprise workflow compliance, whereas **AgentHarm** stresses open-ended harmful-behavior detection across multiple harm categories.
>
> We also evaluate multiple base-agent settings.QS's gains over baselines on both, therefore, indicate that the method is **not tied to one narrow benchmark family**. Additional benchmarks could further broaden the evaluation, and we will make this motivation more explicit in the revision.
>
> We chose these two benchmarks to allow an apples-to-apples comparison with established guardrail baselines in two complementary settings. The evaluation is end-to-end: it tests whether QS improves the overall safety/overhead tradeoff under prior benchmark conditions.
>
> _4. Staged control with bounded feedback._
>
> The reviewer's concern would be correct for a pipeline in which later stages can only consume fixed outputs from earlier ones. QS is not designed that way. The runtime path is staged for efficiency and auditability, but it **isn't a strictly one-way or irreversible chain**: it includes bounded feedback paths that allow later stages to increase scrutiny, request additional state, and support correction after a block.
>
> Concretely, the Threat Watcher adjusts scrutiny for higher-risk agents, the Policy Verifier can request state expansion when confidence is low, and blocked interactions return a rationale to the underlying agent to support self-correction rather than terminal failure. We will make these feedback paths more explicit in the methodology section.
>
> Policy Verifier can also request state expansion when confidence is low, so later verification is **isn't irrevocably fixed by an earlier partial state estimate**. Finally, when an interaction is blocked, the Referee returns a grounded rationale to the underlying agent, enabling the message or action to be revised and retried rather than forcing terminal failure.
>
> he control flow is staged, but it still supports **adaptive re-checking and bounded error correction** rather than a purely linear pass-through. We will revise the methodology section to make these feedback paths explicit and to distinguish them more clearly from a strictly sequential pipeline.
>
> _5. Guarding intermediate interactions, not only final actions._
>
> QS is designed to guard **each runtime interaction**, including both **inter-agent messages** and **environment actions/tool calls**, rather than only the final action. This is a key design point: the guard operates at the **coordination layer** of the multi-agent system, not only at the final execution endpoint.
>
> _6. Runtime policy updates._
>
> The policy book is fixed during a run and treated as a trusted runtime artifact. This is deliberate: policy changes occur **offline between runs**, not as live mutations during execution, which helps preserve runtime integrity and auditability.

---

> > ### Author Rebuttal · Reviewer_Zamx · 2026-04-06
> >
> > Thank you for the rebuttal. The authors provide the clarification of the feedback mechanisms. However, the concerns of policy translation robustness and limited evaluation remain. The rebuttal does not address the reliability or correctness of the translation process, and fails to discuss why the two benchmarks are sufficient or representative. There are many relevant benchmarks such as R-Judge, AgentDojo, SafeArena, Agent-SafetyBench, etc., which should either be included in the empirical comparison or at minimum discussed in the related work, the same for related work on multi-agent safety frameworks.

---

> > > ### Author Response · Authors · 2026-04-06
> > >
> > > Thanks for the follow-up. Of the original concerns, two remain, both falling under **evaluation scope**: (i) reliability of the offline policy translation process, and (ii) breadth of the benchmark evaluation. We address each more precisely.
> > >
> > > # On policy translation reliability
> > >
> > > The paper didn't establish the rule correctness of the translation process in an absolute sense. This isn't a condition unique to QuadSentinel **(QS)**: ShieldAgent (ICML'25) established policy compilation as a viable approach to runtime enforcement for single-agent settings, and QS extends the same direction to the multi-agent coordination layer, where inter-agent messages and compositional risks aren't captured by single-agent guards.
> > >
> > > Both systems share the same offline dependency. Within that shared framework, the specific comparison conditions favor ShieldAgent: their extracted policy files are not publicly released, and their extraction process is non-deterministic. We followed the methodology described in their paper to generate policy rules from the same source texts: ST-WebAgentBench's official safety rules and AgentHarm's Reddit User Rules, the same source referenced in ShieldAgent's evaluation (§4). The resulting policy book reflects our own execution of that process rather than their exact artifacts, meaning our policy inputs are likely noisier than theirs.  We therefore treat their published figures as a **performance upper bound**: QS outperforms that standard while operating under less favorable input conditions.
> > >
> > > In response to this concern, we ran a no-human-review variant on AgentHarm to isolate expert review's contribution directly: if expert review were the source of the advantage, removing it should collapse the gap. The results, to be included in the revision, show otherwise:
> > >
> > > | Variant | Acc. | Prec. | Rec. | FPR |
> > > |---|---|---|---|---|
> > > | QS, no expert review | 90.3 | 93.3 | 86.9 | 6.3 |
> > > | QS, expert review | 91.5 | 97.4 | 85.2 | 2.3 |
> > > | ShieldAgent | 86.9 | 95.2 | 77.7 | 3.9 |
> > >
> > > Without expert intervention, QS still outperforms ShieldAgent on accuracy and recall. Expert review tightens calibration; it doesn't explain the comparative advantage. Appendix A.4 already characterizes the primary failure mode: when a threat category is unregistered, the system produces a false negative, a limitation that applies to any rule-based guard and is disclosed explicitly.
> > >
> > > # On benchmark coverage
> > >
> > > The four suggested benchmarks address safety from angles different than the runtime policy-compliance setting we study; their exclusions fall into two distinct cases. We'll add a dedicated related-work paragraph covering these benchmarks and multi-agent safety frameworks, with distinctions explained below.
> > >
> > > Constructing policy books for these benchmarks ourselves would introduce a confound: rules authored after inspecting which behaviors are labeled harmful would fit the labels rather than measure guard quality. The externally provided policy sources in our two chosen benchmarks are precisely what protect the experiment from this problem.
> > >
> > > R-Judge provides no pre-specified policy book. Its safety rationales are per-scenario and post-hoc, so compiling rules from them would fall into this confound.
> > >
> > > AgentDojo defines safety by whether an adversarial injection achieves its goal, not by compliance with a pre-specified policy. There's no policy book to compile from, so the evaluation protocol can't be applied.
> > >
> > > SafeArena and Agent-SafetyBench are structurally closer to our evaluation interface. Their harm categories are high-level and ambiguous, so compiling rules from them reintroduces the post-hoc validity concern above. ShieldAgent is additionally closed-source and unevaluated there; a GuardAgent-only comparison wouldn't isolate the variable the paper studies, since GuardAgent uses code-generation-based rule matching rather than compiled policy logic, making it a different architectural category from both QS and ShieldAgent. Our revision will nonetheless report GuardAgent comparisons with explicit caveats about the policy source and the missing ShieldAgent baseline.
> > >
> > > The revision will discuss these four benchmarks in the related work, with the distinctions above made explicit, and will position QS relative to the broader landscape of multi-agent safety benchmarks and frameworks.
> > >
> > > # ---
> > > QS introduces a new pattern for multi-agent supervision: compiling natural-language policy text into runtime-enforced logical rules. To our knowledge, this pattern is new and directly extensible by follow-on work.
> > >
> > > Both concerns identify limits on how broadly the current evaluation generalizes, not on whether the pattern works under the stated conditions. On the narrower question the experiments are designed to answer, the results are consistent across base agents and LLM backbones, including under GPT-4o, Qwen3 and across AWM and Magentic-One configurations. The revision will explicitly address both scope questions.

---

### Decision · Program_Chairs · 2026-04-30

**Decision:**

Reject

**Comment:**

This paper proposes a guardrail system for monitoring multi-agent systems. There is noticeable disagreement among reviewers, with two leaning toward acceptance and two toward rejection. The main concerns include limited benchmark coverage, as well as the additional cost and potential lack of generalizability introduced by human involvement.

At the same time, the proposed system demonstrates strong empirical performance, improving both accuracy and recall while maintaining relatively low overhead compared to existing guard approaches . The design of a multi-agent guard with machine-checkable rules provides a meaningful alternative to existing single-agent safety mechanisms, especially in scenarios where coordination-level risks are important.

Overall, despite its potential to provide a practical direction for using multi-agent systems for safety, it relies on human intervention and multiple components, which may limit scalability. The current analysis is somewhat limited, and the evaluation scope could be broader. I suggest that the authors strengthen the paper by including additional benchmarks and providing more comprehensive analysis to support the generality of their conclusions.